

# Development of a wind-based storm surge model for the German Bight

Laura Schaffer[1,2], Andreas Boesch[1], Johanna Baehr[2], and Tim Kruschke[1]

[1]Federal Maritime and Hydrographic Agency (BSH), Hamburg, Germany
[2]Institute of Oceanography, Universität Hamburg, Hamburg, Germany

**Correspondence:** Laura Schaffer (laura.schaffer@bsh.de)

**Abstract.** Storm surges pose significant threats to coastal regions, including the German Bight, where strong winds from north-westerly directions drive extreme water levels. In this study, we present a simple, effective storm surge model for the German Bight, utilizing a multiple linear regression approach based solely on 10 m effective wind as the predictor variable. We train and evaluate the model using historical skew surge data from 1959 to 2022, incorporating regularization techniques to improve prediction accuracy while maintaining simplicity. The model consists of only five terms, namely the effective wind at various locations with different time lags within the North Sea region, and an intercept. It demonstrates high predictive skill, achieving a correlation of 0.882. This indicates that, despite its extreme simplicity, the model performs just as well as more complex models. The storm surge model provides robust predictions for both moderate and extreme storm surge events. Moreover, due to its simplicity, the model can be effectively used in climate simulations in future studies, making it a valuable tool for assessing future storm surge risks under changing climate conditions, independent of the ongoing and continuous sea level rise.

## 1 Introduction

Many of the world's coasts are endangered by storm surges. These can have devastating consequences causing widespread destruction and even loss of life (von Storch, 2014). The German Bight, located in the south-eastern part of the North Sea (Fig. 1), is an example of a region that is prone to frequent and severe storm surges. The major driver of storm surges in the German Bight is strong wind from north-westerly directions which are associated with intense extra-tropical cyclones travelling from the North Atlantic into the North Sea region. The co-occurrence of such storms with high tides leads to high coastal water levels, potentially resulting in flooding, erosion, and significant damage to infrastructure. Coastal protection institutions have been confronted with the challenge of managing these sudden extreme water level events for decades. The continuing rise of sea levels induced by anthropogenic climate change adds even greater urgency to this issue. Assessing the potential of an impending storm surge is therefore important in order to ensure public safety and maintain regional infrastructure.

In addition to other factors such as sea level pressure and external surges (Böhme et al., 2023), wind proves to be the primary driver for sea level variability in the German Bight (Dangendorf et al., 2013) and therefore plays an important role in estimating



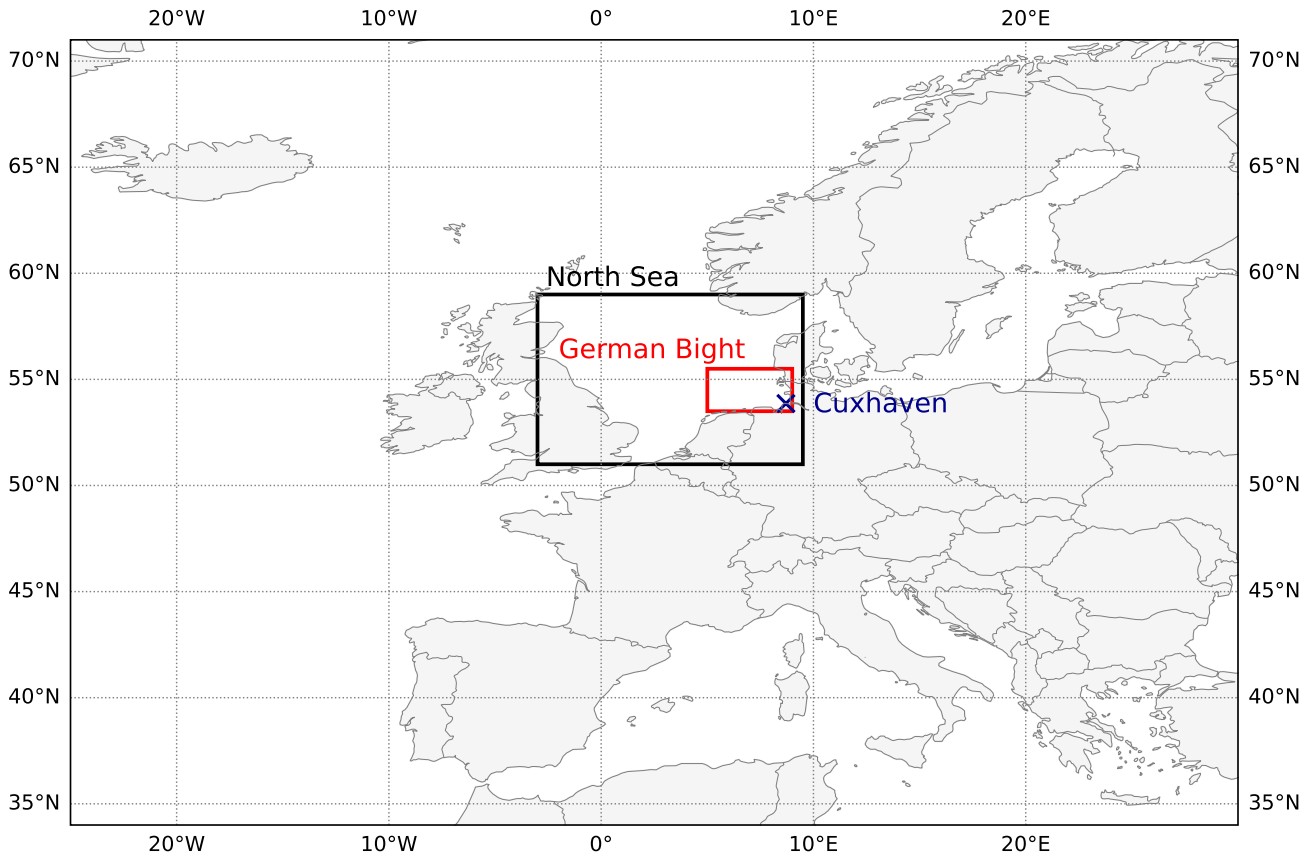

**Figure 1.** Map with marked areas of the North Sea (black box) and German Bight (red box). The blue cross indicates the location of Cuxhaven.

the height of storm surges. Various studies investigate the meteorological conditions that caused extreme storm surges in the past. While they focus on just a few case studies, they demonstrate the successful applicability of reanalysis data for recreating these events: The storm surges studied include, for example, the well known Hamburg storm surge of 1962 (Jochner et al., 2013; Meyer and Gaslikova, 2024), one of the highest storm surges ever measured in 1976 (Meyer and Gaslikova, 2024)

and the storm surge caused by storm 'Xaver' in 2013 (Dangendorf et al., 2016; Meyer and Gaslikova, 2024). While these and other studies focus on the analysis of individual historical events, Krieger et al. (2020) concentrate on the statistics and long-term evolution of German Bight storm activity. They find that storm activity over the German Bight is characterized by multidecadal variability. This is in line with the results of Dangendorf et al. (2014), who find that storm surges in the North Sea are characterized by inter-annual to decadal variability associated with large-scale atmospheric circulation patterns.

Nevertheless, in order to estimate the resulting storm surge levels in the German Bight, a translation of wind speed into water level is required. Particularly, the skew surge, the difference between the observed and predicted high (or low) water within a



tidal cycle, provide reliable information about storm surges (Williams et al., 2016; Ganske et al., 2018). Two approaches are commonly employed to achieve this translation: one uses the so-called effective wind as a proxy, while the second approach involves predicting skew surges using a statistical model.

For the former, a variable - the effective wind - consisting of wind speed and direction is used to objectively assess the conditions for a storm surge in the German Bight (Müller-Navarra et al., 2003; Jensen et al., 2006). Cuxhaven (Fig. 1), located in the center of the German Bight coast and with its long data series, is often used as a proxy for that region. For Cuxhaven, the effective wind is defined as the fraction of the 10 m wind blowing from direction 295°. In an empirical study, this wind direction was determined as the one for which the wind-induced increase of water levels in the German Bight is greatest (Müller-Navarra

et al., 2003; Jensen et al., 2006). Averaged over the German Bight, the effective wind is therefore a measure of the contribution of wind to storm surges at the German Bight coastline and can be used as an indicator for the storm surge potential of a weather situation (Jensen et al., 2006). Befort et al. (2015) use the effective wind in combination with a storm tracking algorithm to detect storm surges in the German Bight. Using this method, they show an improvement in the identification of storm surge events compared to the sole use of the effective wind. This agrees with findings by Ganske et al. (2018), who state that a high

effective wind alone is not necessarily linked to a large storm surge height, but that additional parameters such as the storm track need to be taken into account.

Building upon the second approach, namely the use of a statistical model, Müller-Navarra and Giese (1999) re-examined and revised existing studies and similar methods from the 1960s, setting up a statistical model using multiple linear regression in order to determine storm surge situations in the German Bight, i.e. in Cuxhaven. As predictors for computing the skew surge

in Cuxhaven, they use wind speed and direction, air and sea surface temperature, air pressure and its 3-hourly change, water level in Wick on the Scottish east coast 12 h earlier and water levels in Cuxhaven during the immediately preceding low and high waters. They find that the external surge and auto-correlation improve the model performance. The result is an empirical model with 14 basic functions that is able to describe the skew surge height in Cuxhaven (Müller-Navarra and Giese, 1999). Several other studies built on this study and base their models on the mathematical approaches by Müller-Navarra and Giese

(1999). Jensen et al. (2013), for example, successfully examine the empirical-statistical relationship between wind, air pressure and the skew surge in Cuxhaven for the period 1918 to 2008. As an approximation for external surges, they use air pressure and wind time series at the northern edge of the North Sea. Dangendorf et al. (2014) extend this analysis by reconstructing storm surges in the German Bight back to 1871, but with higher predictive skill from 1910 onwards. However, it is worth noting that their model is only based on atmospheric surface forcing as predictors - external surges are excluded in the model setup

(Dangendorf et al., 2014). Niehüser et al. (2018) apply a similar model setup, but for multiple tide gauge locations along the North Sea coast, focusing on the period from 2000 to 2014. In contrast to using wind and air pressure information of the nearest grid cell for each tide gauge location (Jensen et al., 2013; Dangendorf et al., 2014), Niehüser et al. (2018) implement a step wise regression approach, considering time lags up to 24 h. This allows them to determine the spatial and temporal positions of relevant predictors for each site. Their model shows comparable skill measures to the more complex model by Jensen et al.

70 (2013).

However, as all these statistical modeling approaches require a large number of input variables for very specific locations or



regions, most of them have been employed solely based on observational data or atmospheric reanalysis (Müller-Navarra and Giese, 1999; Jensen et al., 2013; Dangendorf et al., 2014; Niehüser et al., 2018). In the context of climate change, the assessment of future storm surge risk is of major importance (IPCC, 2023). In order to incorporate necessary climate projections, a less

complex statistical model is needed that is applicable to a multi-model ensemble of climate model simulations with only a limited number of variables available and a comparatively coarse spatial resolution.

To fill this gap, this paper seeks to (i) set up a simple storm surge model for Cuxhaven using a multiple linear regression approach based only on the 10 m wind from the ERA5 reanalysis as predictor variable; (ii) improve prediction accuracy and reduce model complexity by applying regularization methods; and (iii) assess the model's performance by using cross-

validation methods and classification evaluation. The reason for the restriction to winds is that most climate model simulations provide corresponding information. Thus, this paper introduces a new method for predicting storm surge heights in the German Bight, which can also be applied using data from climate model projections in future studies.

## 2   Methods and data

We develop a simple storm surge model for the German Bight, i.e. Cuxhaven, based on a multiple linear regression approach

and relying exclusively on the grid cell specific 10 m effective wind as predictor variable. Broadly speaking we follow four steps (Fig. 2): (a) data pre-processing (Sect. 2.1.1 and 2.1.2); (b) the identification of relevant spatial and temporal positions of the predictor across the entire North Sea region (namely certain grid cells of the atmospheric reanalysis used; Sect. 2.2.1 and 3.1); (c) choosing an appropriate threshold value and regularization method for training the model (Sect. 2.3.1, 3.2 and 3.3); and (d) training and evaluation of the storm surge model (Sect. 2.3.2, 2.3.3, 3.4 and 3.5).

### 2.1   Data

#### 2.1.1   Skew surge data

A valuable metric for storm surge research is the skew surge at high waters (de Vries et al., 1995; Jensen et al., 2013; Williams et al., 2016). It is the height difference between the highest recorded sea level and the predicted tidal high water within a tidal cycle, regardless of the time. We create a time series of high water skew surge (*hwss*) at the tide gauge Cuxhaven for the years

1959 to 2022. In order to derive *hwss*, we calculate consistent tidal predictions for these years based on the observed times and water levels of high waters. The observation data is collected and quality-checked by the Waterways and Shipping Office Elbe-Nordsee, which operates the Cuxhaven tide gauge. The height reference is tide gauge zero.

We perform tidal analysis and prediction using the method of "harmonic representation of inequalities". We opt for this method for two reasons: 1) It is a proven operational technique that delivers excellent results under the tidal conditions in the German

Bight (Horn, 1960; Boesch and Müller-Navarra, 2019; Boesch and Jandt-Scheelke, 2020), and 2) this technique is applied directly to the heights of the vertices which are needed for this study. For the tidal analysis we follow the scheme used for the official tide tables: the predictions for a year are based on a tidal analysis of the 19 water level observation years ending three





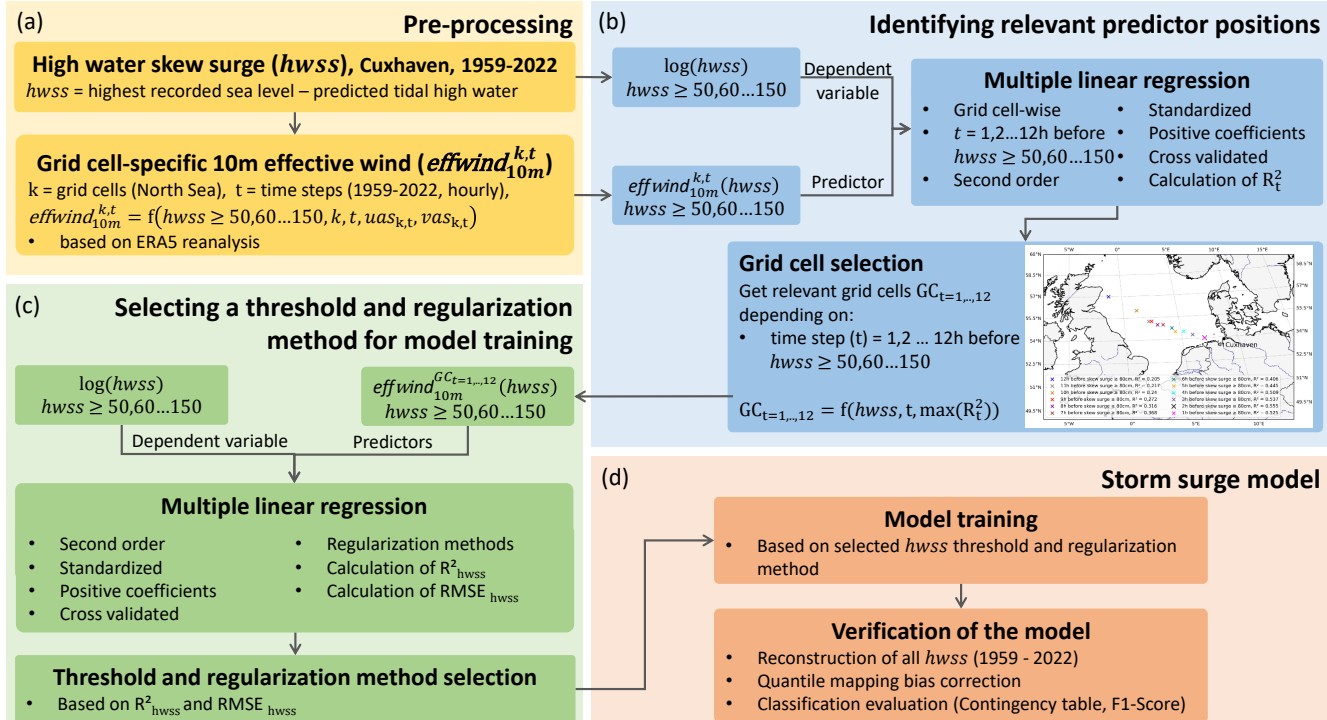

**Figure 2.** Schematic representation of the process for developing the wind-based storm surge model. (a) shows the data pre-processing steps; (b) shows the identification of relevant predictor positions; (c) shows the procedure for selecting a threshold regularization method for model training; (d) shows the final model training and evaluation.

years before the year of the prediction. For example, the prediction for the year 1959 is based on the results of the tidal analysis of the observations from 1938 to 1956. The tidal analysis employs 39 long-period tidal constituents and runs in two iterations, with a three-sigma clipping of outliers before each iteration. We derive the skew surge for 45162 high waters compared to 45169 semi-diurnal tidal cycles theoretically present in the studied years. It is worth noting, that the skew surge data does not take into account sea level rise.

**Table 1.** Number of data points for each sub-sample based on the selected threshold.

| hwss $\geq$ | 50 cm | 60 cm | 70 cm | 80 cm | 90 cm | 100 cm | 110 cm | 120 cm | 130 cm | 140 cm | 150 cm |
|---|---|---|---|---|---|---|---|---|---|---|---|
| Sample size | 3761 | 2730 | 2018 | 1504 | 1161 | 923 | 717 | 551 | 431 | 329 | 253 |

As this study aims for a storm surge model we only use a sub-sample of high *hwss* events for training. As part of this study, we test various thresholds as a lower boundary for defining this sub-sample, specifically ranging from 50 to 150 cm. We decided





on this range as it represents a fair compromise between data availability and the official storm surge definition of greater than or equal to 150 cm above mean high water (Sect. 2.3.3). In the following we refer to this range as *hwss* $\geq 50, 60...150$ cm. We show the number of available data points for each sub-sample in Table 1. We describe the choice of the final threshold in Sect. 2.3.1 and 3.2. As the selected skew surge range is not normally distributed, we apply the natural logarithm to normalize the data set before using it. Consequently, we train the model based on logarithmic values.

### 2.1.2   Grid cell-specific effective wind

The effective wind refers to the combined effect of wind speed and direction. It describes the proportion of the wind projected onto a specific wind direction. Here, the specific wind direction is the one that causes a certain wind-related water level in Cuxhaven. We determine the specific wind direction separately for each *hwss* training threshold (*hwss* $\geq 50, 60...150$ cm). Therefore, we carry out a grid cell wise composite analysis of the corresponding zonal (*uas*) and meridional (*vas*) components

of the 10 m wind from the ERA5 reanalysis.

ERA5, generated by the European Centre of Medium-Range Weather Forecasts (ECMWF), is the most recent climate reanalysis offering hourly data on various atmospheric, land-surface, and sea-state variables. The data is provided at a horizontal resolution of 31 km and 137 levels in the vertical, covering the period from 1940 onwards (Hersbach et al., 2020).

As we started our analysis when the backward extension to 1940 was not yet available, we only use the hourly wind compo-

nents for the period 1959 to 2022. Since our study area is the German Bight, we focus exclusively on the grid cells in the North Sea region, which we define from -5° E to 10.5° E in longitude and from 51° N to 59° N in latitude. For this region, we count 2079 grid cells. For each of these grid cells ($k$) and in hourly time steps ($t$) we calculate the effective wind separately for each *hwss* training threshold as follows:

$$effwind_{10\mathrm{m}}^{k,t}\left(hwss\right) = \left(\frac{\overline{uas}_k}{\overline{U}_k}\right) \cdot uas_{k,t} + \left(\frac{\overline{vas}_k}{\overline{U}_k}\right) \cdot vas_{k,t}. \tag{1}$$

First, we normalize the mean zonal ($\overline{uas}_k$) and meridional ($\overline{vas}_k$) wind components from the composite analysis by dividing each by the mean wind speed ($\overline{U}_k$). We subsequently calculate the effective wind ($effwind_{10\mathrm{m}}^{k,t}\left(hwss\right)$) by projecting the actual wind components ($uas_{k,t}$ and $vas_{k,t}$) onto the corresponding normalized mean components ($\frac{\overline{uas}_k}{\overline{U}_k}$ and $\frac{\overline{vas}_k}{\overline{U}_k}$). The result is the effective wind for the period 1959 to 2022 in hourly time steps and individually for each grid cell in the North Sea region. The particular feature of $effwind_{10\mathrm{m}}^{k,t}\left(hwss\right)$ is the fact that its value can be negative. This is the case as soon as the wind blows

from the opposite direction to the specific wind direction.

## 2.2   Multiple linear regression approach

### 2.2.1   Identifying relevant predictor positions

Similar to Niehüser et al. (2018), but using a different method, we apply the approach of non-static predictors. This method only considers predictor locations that are relevant for the observed skew surge variability in Cuxhaven. Thus, instead of just

considering the nearest grid cell for the Cuxhaven gauge (Jensen et al., 2013; Dangendorf et al., 2014), we identify grid cells and




time lags across the entire North Sea region that are most relevant for the respective sub-sample ($hwss \geq 50, 60...150$ cm). In a first step, we select all time steps of $effwind_{10\text{m}}^{k,t}$ ($hwss$) 12 hours before the respective occurrence of $hwss \geq 50, 60...150$ cm in Cuxhaven. Since $effwind_{10\text{m}}^{k,t}$ ($hwss$) is available in hourly resolution, this corresponds to 12 time steps. We choose 12 hours, as this is approximately the time between two high tides.

Subsequently, we create an individual model for each grid cell in the North Sea region at each time step using multiple linear regression. In the following, we refer to these models as grid cell models. Since we count 2079 grid cells in the North Sea region and consider 12 time steps, we have a total of 24948 grid cell models for each of the 11 $hwss$ training threshold. The dependent variable in each model is the respective logarithmic $hwss$ height, with the standardized $effwind_{10\text{m}}^{k,t}$ ($hwss$) serving as predictor. As the wind stress depends on the square of the wind speed (Olbers et al., 2012), we create the grid cell models with quadratic order. In addition, we force the grid cell models to have exclusively positive coefficients. We justify this by the use of the effective wind as predictor variable: In order to take into account the resulting physical consequence of negative effective winds, namely a water movement away from the coast of Cuxhaven, we set up the grid cell models with only positive coefficients. In this way, we ensure the effect of the negative sign.

Once we have defined the conditions for the grid cell models, we perform a leave-one-out-cross-validation and calculate the coefficient of determination ($R^2$) for each grid cell model. In order to obtain the most relevant grid cells for the respective sub-sample of events, we select the grid cell model with the highest $R^2$ from each time step. As we consider the 12 hours before the occurrence of $hwss \geq 50, 60...150$ cm, we obtain 12 grid cell models and thus grid cells ($GC_{t=1,..,12}$), each of which contains both the relevant time lag and position in the North Sea region for the respective $hwss$ training threshold in Cuxhaven (Fig. 2, b).

### 2.2.2 Regularization methods

The usual regression procedure for determining the unknown coefficients in a multiple linear regression model is based on the Ordinary Least Squares (OLS) method. Its error criterion is the minimization of the sum of the squared errors, where the error is the difference between the actual and the predicted value (Wilks, 2011). However, OLS has some known disadvantages. For example, OLS is highly sensitive to outliers in the data and can be prone to overfitting. In addition, OLS performs poorly when it comes to accuracy in predicting unseen data and in interpreting the model (Zou and Hastie, 2005). For the latter, the simpler the model, the more the relationship between the dependent variable and the predictors is highlighted. The model's simplicity is particularly important when the number of predictors is large.

An effective method for overcoming the difficulties of OLS is regularization. It is a method to avoid overfitting and control the complexity of models by adding a penalty term to the model's target function during training. The aim is to keep the model from fitting too closely to the training data and to encourage simpler models that are better suited to unseen data. Here, we consider three different regularization techniques, namely ridge (Hoerl and Kennard, 1970a, b), lasso (Tibshirani, 1996) and elastic net regression (Zou and Hastie, 2005).

The main idea behind ridge regression (Hoerl and Kennard, 1970a, b) is its ability to strike a balance between bias and variance. This is achieved by including a penalty term proportional to the square of the coefficients in the loss function.





This approach leads to the coefficient estimates being smaller, while still having non-zero values. The target function to be minimized becomes:

$$\text{Loss}_{\text{Ridge}} = \text{Loss}_{\text{OLS}} + \lambda \sum_{j=1}^{n} \beta_j^2, \tag{2}$$

where $\text{Loss}_{\text{OLS}}$ is the loss function of Ordinary Least Squares, $\lambda$ is the regularization parameter that defines the regularization strength and $\sum_{j=1}^{n} \beta_j^2$ is the regularization term - the sum of squared coefficients. Since large coefficients may lead to low bias,

but high variance, they are penalized by adding the regularization term and effectively shrink towards zero. Consequently the model becomes sensitive to fluctuations in the data. However, ridge regression is unable to create a simple model, in the sense of fewer predictors, as it always retains all predictors in the model. This contrasts with the lasso regression (Tibshirani, 1996). Lasso regression is a method to reduce overfitting and perform predictor selection by setting some coefficient estimates to exactly zero. For this purpose, a regularization term proportional to the absolute value of the coefficients is added to the loss

function. The target function to be minimized becomes:

$$\text{Loss}_{\text{Lasso}} = \text{Loss}_{\text{OLS}} + \lambda \sum_{j=1}^{n} \mid \beta_j \mid, \tag{3}$$

where $\text{Loss}_{\text{OLS}}$ is the OLS loss function, $\lambda$ is the regularization parameter and $\sum_{j=1}^{n} \mid \beta_j \mid$ is the regularization term - the sum of absolute values of coefficients. Lasso regression simplifies the model by shrinking less important coefficients to zero, effectively eliminating some predictors from the model. This leads to simpler and more interpretable models.

Zou and Hastie (2005) propose a third regularization method whose main principle is to balance ridge and lasso regression by adding both ridge and lasso regularization terms to the loss function. The target function to be minimized becomes:

$$\text{Loss}_{\text{Elastic Net}} = \text{Loss}_{\text{OLS}} + \lambda_1 \sum_{j=1}^{n} \mid \beta_j \mid + \lambda_2 \sum_{j=1}^{n} \beta_j^2, \tag{4}$$

where $\text{Loss}_{\text{OLS}}$ is the loss function of OLS, $\lambda_1$ and $\lambda_2$ are the regularization parameters for lasso and ridge penalties respectively, $\sum_{j=1}^{n} \mid \beta_j \mid$ is the lasso regularization term and $\sum_{j=1}^{n} \beta_j^2$ is the ridge regularization term. Elastic net regression combines

the strength of both lasso and ridge regression by balancing the selection of predictors and coefficient shrinkage, resulting in more robust and reliable predictive models.

### 2.2.3  Model development

We apply all three regularization techniques (Sect. 2.2.2) separately for every sub-sample ($hwss \geq 50, 60...150$ cm) and per-

form multiple linear regression using ridge, lasso and elastic net techniques. As we count 11 different $hwss$ training thresholds and three regularization methods, we arrive at 33 models. In the following, we refer to these models as skew surge models. For each skew surge model, we use the effective wind of the 12 relevant grid cells ($effwind_{10\text{m}}^{GC_{t=1,...,12}} (hwss)$) identified earlier (Sect. 2.2.1) as predictor variables. We stress that each of the 12 predictor variables represents a specific position and a time lag





in the North Sea region. As before, we use the standardized $effwind_{10m}^{GC_{t=1,...,12}}(hwss)$ and set up the skew surge models with

identical conditions as the grid cell models. That is, in quadratic order and with forced positive coefficients. In addition, we determine the respective regularization parameter $\lambda$ for ridge, lasso and elastic net regression for each skew surge model using cross-validation. Subsequently, we perform a leave-one-out-cross-validation for each skew surge model and regularization technique (Fig. 2, c).

### 2.3 Evaluation and setup of the storm surge model

#### 2.3.1 Selecting a threshold and regularization method for model training

In order to evaluate the skew surge models, we calculate $R^2$ and the Root-Mean-Square-Error (RMSE). RMSE is a measure of how well the model is performing in terms of prediction accuracy, with lower values indicating better performance. It has the same units as the observed values - in this case cm. We also determine the 95% confidence interval for the RMSE by applying the bootstrap method. This method involves resampling from the original dataset - here 1000 times - with replacement to

simulate multiple scenarios. Finally, based on $R^2$, RMSE, data availability and the ability of the individual skew surge models to capture even very severe storm surges, we decide on one final *hwss* threshold for model training (Sect. 3.2) and for one regularization method (Sect. 3.3). The model trained on the selected *hwss* sub-sample using the chosen regularization method is hereafter referred to as the storm surge model.

#### 2.3.2 Final storm surge model setup with quantile mapping bias correction

We evaluate the storm surge model by predicting all *hwss*, that is every 12 hours, for the years from 1959 to 2022. In doing so, we exclude the year to be predicted from the training. As before, we train the storm surge model based on the sub-sample corresponding to the previously selected *hwss* event threshold. Here, the predictors, namely $effwind_{10m}^{GC_{t=1,...,12}}(hwss)$, are mostly positive, as they correspond to the wind direction relevant for the *hwss* sub-sample. For the year that the model is supposed to predict, we have to take all high tides into account. In doing so, we come across skew surge heights that are smaller than the

skew surge height with which the model was trained, or even negative. Smaller or sometimes negative *hwss* indicate a minor water movement or even a water movement away from the coast of Cuxhaven. Here, $effwind_{10m}^{GC_{t=1,...,12}}(hwss)$, which is mainly responsible for this water movement, tends to take on negative values. Since the storm surge model consists of interaction terms and squared terms, negative $effwind_{10m}^{GC_{t=1,...,12}}(hwss)$ as predictors would still lead to positive values. This means that despite negative $effwind_{10m}^{GC_{t=1,...,12}}(hwss)$, the model would predict positive *hwss*, which is contrary to the physical consequence of

negative $effwind_{10m}^{GC_{t=1,...,12}}(hwss)$. To overcome this problem, we create a mathematical condition that we apply solely to the dataset of the year to be predicted. This mathematical condition specifies that whenever a predictor is negative, the coefficient associated with its squared term becomes negative. Furthermore, if both predictors in an interaction term are negative, the coefficient before the term becomes negative here too. This mathematical condition, combined with the forced positive coefficients in the training, allows the model to reproduce the physical consequence of negative $effwind_{10m}^{GC_{t=1,...,12}}(hwss)$.

However, since we train the model on logarithmic *hwss*, the model will always predict *hwss* greater than zero. In order to





address this issue, we apply the quantile mapping bias correction technique based on the equations of Cannon et al. (2015). With this method we aim to minimize distributional biases between predicted and observed *hwss* time series. Its interval-independent approach considers the entire time series, redistributing predicted values based on the distributions of the observed *hwss* (Fig. 2, d).

### 2.3.3 Classification evaluation

We perform a classification evaluation to investigate whether the model correctly classifies the predicted storm surges according to the storm surge definition for the German North Sea coast. The latter is defined by the height of the water level above mean high water (MHW). According to the definition of the Federal Maritime and Hydrographic Agency (BSH), a storm surge event along the German North Sea coast is classified as follows: an increase in water level of 150 to 250 cm above MHW is referred to as a "storm surge"; if the water level reaches between 250 to 350 cm above MHW, it is classified as a "severe storm surge"; and any event exceeding 350 cm above MHW is referred to as a "very severe storm surge" (Müller-Navarra et al., 2012).

To assess the extent to which the storm surge model assigns the predicted *hwss* to the correct classes, we calculate the F1 score (van Rijsbergen, 1979). The F1 score combines precision and recall into one metric. Precision measures the accuracy of positive predictions made by the model. It is determined by dividing the number of true positive predictions by the total number of samples predicted as positive, regardless of correct identification:

$$Precision = \frac{TP}{TP+FP}, \tag{5}$$

where $TP$ represents true positive predictions, while $FP$ indicates false positive predictions. Recall measures the fraction of actual positives that were correctly identified by the model. It is computed by dividing the number of true positive predictions by the total number of samples that should have been identified as positive:

$$Recall = \frac{TP}{TP+FN}, \tag{6}$$

where $TP$ refers to true positive predictions and $FN$ to false negative predictions. The F1 score is defined as the harmonic mean of precision and recall:

$$F1 = 2\frac{PR}{P+R}, \tag{7}$$

where $P$ stands for precision and $R$ for recall. The F1 score has its best value at 1 (perfect precision and recall) and its worst at 0. We calculate the F1 score for the classes starting from a storm surge ($\geq 150$ cm) and from a severe storm surge ($\geq 250$ cm) for the predicted and bias-corrected high water skew surges in the years 1959 to 2022.

## 3 Results

### 3.1 Identifying relevant predictor positions

In the first step for the development of a storm surge model, we determine the spatial and temporal positions of the predictors ($effwind_{10\mathrm{m}}^{k,t}(hwss)$) in the North Sea region, individually for each *hwss* training threshold ($hwss \geq 50, 60...150$ cm). For this



purpose, we identify the grid cells with the highest $R^2$ value from the grid cell models we created for each time step (Fig. A1). Fig. 3 shows the spatial and temporal positions of the predictors for the training threshold of $hwss \geq 80$ cm. 12 hours before

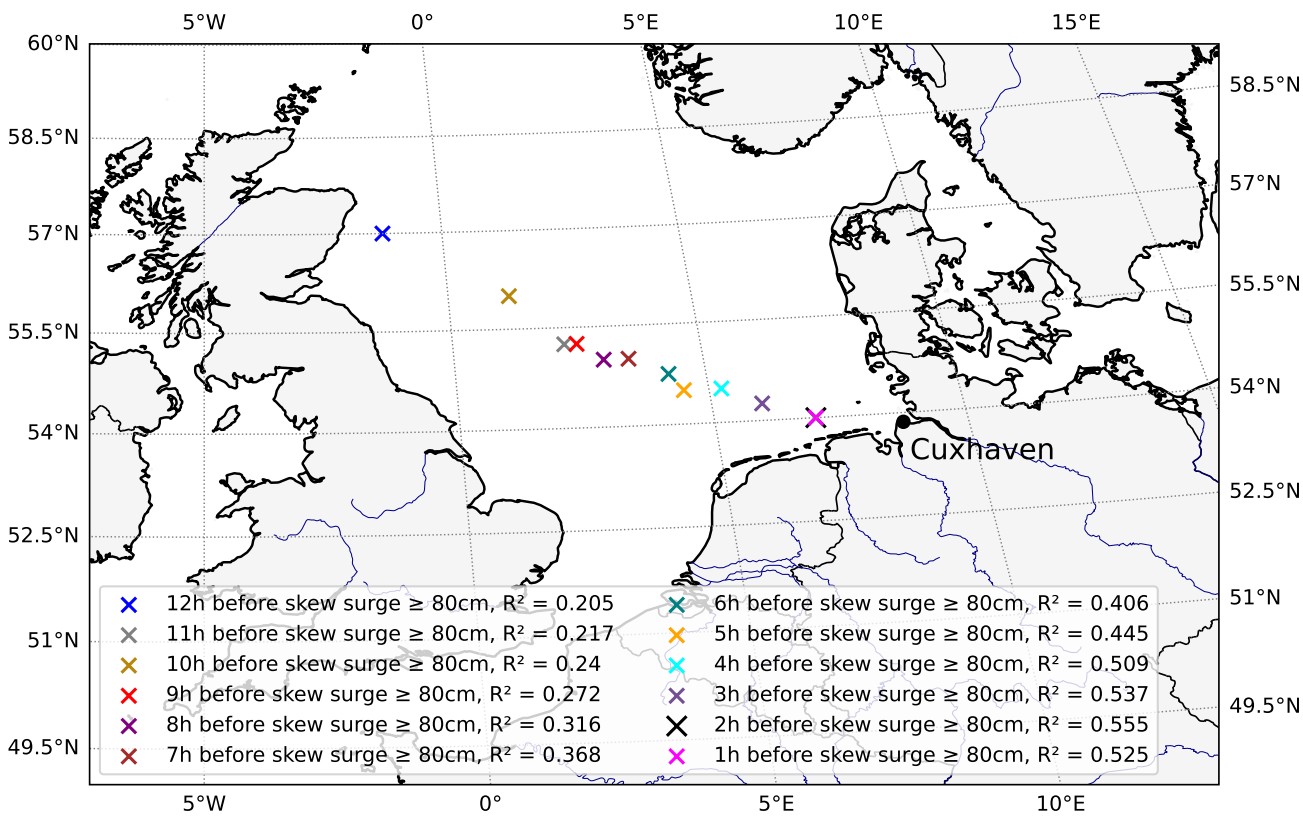

**Figure 3.** Map of the North Sea region showing the 12 relevant grid cells highlighted with crosses. Each cross represents a specific grid cell, with the colour variations indicating the different time intervals before the skew surge event and the corresponding $R^2$ value (based on a leave-one-out-cross-validation). The threshold for training is $hwss \geq 80$ cm.

skew surge events in Cuxhaven, the effective wind in a grid cell in the northwestern North Sea close to Scotland shows the highest explained variance ($R^2 = 0.205$) (blue cross in Fig. 3). With increasing temporal proximity to the skew surge events, the relevance shifts towards the grid cells in the southeastern North Sea. Here, the effective wind in the most southeasterly grid cell, approximately 2 hours ($R^2 = 0.555$) and 1 hour ($R^2 = 0.525$) before the skew surge events, provides the best description of the resulting water level in Cuxhaven (black and pink cross in Fig. 3).

Even though the positions in the North Sea region, i.e. the individual grid cells, vary when applying the different *hwss* subsamples (not shown), the track consistently follows a northwest to southeast orientation. This result is in line with findings





of Meyer and Gaslikova (2024) and Gerber et al. (2016), who investigate several storm surges and conclude that the northern storm tracks induce high surges across the southern North Sea.

## 3.2 Selecting a threshold for model training

Once we have determined the spatial and temporal positions of the predictors (Sect. 3.1), we create skew surge models by applying the three regularization methods. We perform a leave-one-out-cross-validation for the models based on ridge, lasso and elastic net regression separately for the training thresholds $hwss \geq 50, 60...150$ cm, resulting in three models per $hwss$ training threshold. We assess the performance of these skew surge models by computing $R^2$ and RMSE (Fig. 4). We find a general trend of decreasing $R^2$ values across all three models as the training threshold increases, with the highest $R^2$ value of 0.646 ($hwss \geq 50$ cm) and the lowest being 0.569 ($hwss \geq 150$ cm) (Fig. 4, a). Furthermore, skew surge models using ridge regression have a slightly lower $R^2$ value in most cases compared to models using lasso and elastic net regression. Skew surge models based on lasso and elastic net regression show almost equal $R^2$ values across all $hwss$ training thresholds (Fig. 4, a). This trend is also reflected in the RMSE. When trained with lower $hwss$ thresholds, the RMSE is low and the confidence interval is narrow. Conversely, training with higher $hwss$ thresholds results in higher RMSE values and wider confidence intervals. Despite lower $R^2$ values for models with ridge regression, we find no significant differences in performance compared to the other regularization methods trained on the same threshold. We see this in the fact that the confidence intervals of the models trained with the same $hwss$ threshold value overlap regardless of the regularization method. The similarity in performance of the three models, each with a different regularization method, is the result of forcing the positive coefficients. By forcing positive coefficients, ridge regression reduces certain coefficients to zero if their impact on the accuracy of the prediction is small. This process is similar to predictor selection and results in model behaviour similar to lasso and elastic net regression. Consequently, in our case, the performance of ridge regression is closely related to lasso and elastic net regression.

However, when comparing skew surge models across training thresholds, we find that some models trained with higher thresholds (e.g. $hwss \geq 110$ cm) perform significantly worse compared to those trained with lower thresholds (e.g. $hwss \geq 50$ cm) (Fig. 4, b). Additionally, we observe strong fluctuations in the decrease of $R^2$ values and the increase in RMSE values, which become visible from a $hwss$ training threshold of $hwss \geq 90$ cm (Fig. 4). One explanation for this phenomenon and the partly resulting significant differences in model performance might be the availability of data. As the training threshold increases, the amount of available data decreases. Having sufficient data is essential for the model to learn patterns accurately and ensure better performance on unseen data.

In our pursuit of developing a storm surge model capable of simulating extreme events, we aim to train it using the highest possible $hwss$ threshold. However, when selecting from the number of possible training thresholds, we face the challenge of striking a balance between data availability, high threshold and model performance. We decide to use a $hwss$ training threshold of greater than or equal to 80 cm. We base our decision on the following reasons: 1) this training threshold does not lead to a significant difference in model performance compared to other thresholds (Fig. 4, b), 2) we have sufficient data to effectively train the model with this threshold, and 3) we find a satisfactory level of accuracy in simulating extreme events (Fig. 5).





**Figure 4.** $R^2$ (a) and RMSE (b) with the corresponding 95 % confidence intervals of the skew surge models including the different regularization methods across different hwss thresholds for training. The training thresholds are shown on the x-axis with the number of available data points in brackets. Each cross represents a specific regularization method (green: ridge regression, purple: lasso regression, orange: elastic net regression). The results are based on a leave-one-out-cross-validation.

## 3.3 Selecting a regularization method

Following the selecting of a *hwss* training threshold, the next step is to choose a suitable regularization method. Fig. 5 compares the observed and predicted skew surge heights resulting from a leave-one-out-cross-validation using ridge, lasso and elastic net regression, all related to a training threshold of greater than or equal to 80 cm. As mentioned earlier (Sect. 3.2), we observe a similarity in model performance. This is also evident when predicting extremes, with most of the extreme values being



underestimated. This could be attributed to the limited data on extreme skew surge events. Moreover, the coarse resolution of
the atmospheric forcing (ERA5) may contribute to the underestimation of the most extreme events (Dangendorf et al., 2014;
Harter et al., 2024). However, even with a relatively small training threshold ($hwss \geq 80$ cm), we still get an appropriate
representation of the most extreme events (Fig. 5). The problem of underestimating extreme events is not unique to this study.
Several other studies attempting to reconstruct extreme storm surges using statistical models also encounter similar issues
(Dangendorf et al., 2014; Niehüser et al., 2018; Harter et al., 2024).

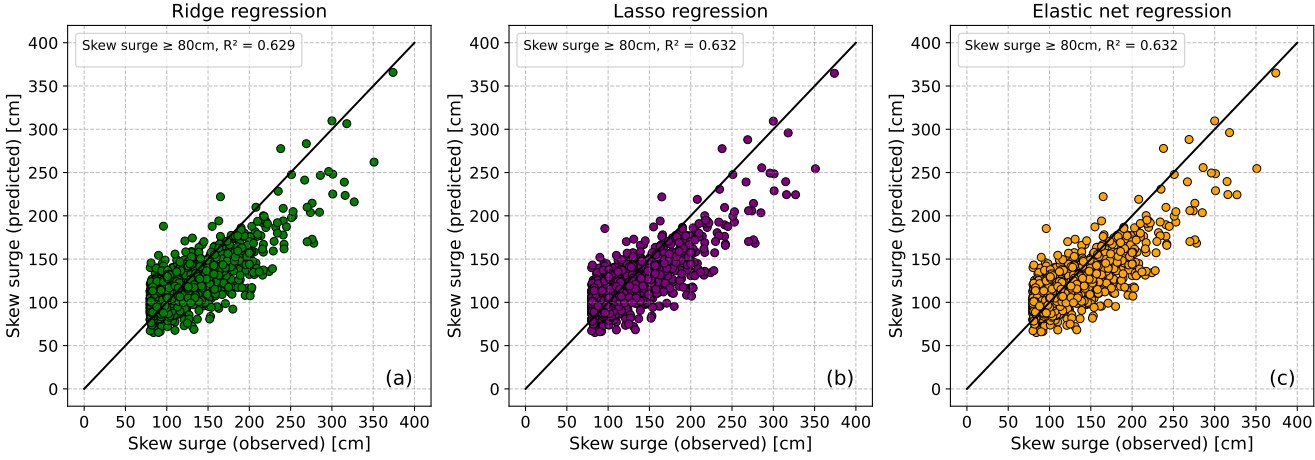

**Figure 5.** Comparison between observed skew surge heights and predicted skew surge heights resulting from a leave-one-out-cross-validation
using ridge (a), lasso (b) and elastic net regression (c). The black line represents the diagonal. The threshold for training is $hwss \geq 80$ cm.

Furthermore, with regard to the performance of the skew surge models, it is important to evaluate the influence of the pre-
dictors. As mentioned above (Sect. 3.2), due to forced positive coefficients, ridge regression performs a lasso-like predictor
selection, resulting in a sparse model with only seven terms including the intercept. However, lasso and elastic net regression
models consist of just five terms in total, including the intercept. Both models show similar performance with identical selected
predictors but slightly different coefficients.

In summary, in our case, we observe an overall high level of predictive performance with sparse models. Considering the three
potential regression methods, we opt for the elastic net regression model (training threshold $hwss \geq 80$ cm) as our final storm
surge model (Fig. 5, c):

$$hwss_{StormSurgeModel} = f\left(effwind_{10\mathrm{m}}^{GC_{t=-12}}, effwind_{10\mathrm{m}}^{GC_{t=-6}}, effwind_{10\mathrm{m}}^{GC_{t=-2}}, effwind_{10\mathrm{m}}^{GC_{t=-1}}\right) + intercept. \tag{8}$$

The *hwss* predicted by the storm surge model is based on the following predictors:

– $effwind_{10\mathrm{m}}^{GC_{t=-12}}$: effective wind in the respective grid cell 12 hours prior the skew surge event (squared),

  – $effwind_{10\mathrm{m}}^{GC_{t=-6}}$: effective wind in the respective grid cell 6 hours prior the skew surge event (squared),



- $effwind_{10\mathrm{m}}^{GC_{t=-2}}$: effective wind in the respective grid cell 2 hours prior the skew surge event (squared),

- $effwind_{10\mathrm{m}}^{GC_{t=-1}}$: effective wind in the respective grid cell 1 hour prior the skew surge event (squared).

This choice is primarily motivated by its simplicity, as it has even fewer terms compared to the ridge regression model, and unlike the lasso regression model, also includes coefficient shrinkage.

### 3.4 Verification of the storm surge model and quantile mapping bias correction

We eventually verify the storm surge model by using it to predict all high water skew surge events from 1959 to 2022. By including this broader range of events, we ensure that the model is capable of predicting skew surges on which it was not trained, namely those below 80 cm. In doing so, we train the model based on all years except the year we want to predict

(Sect. 2.3.2).

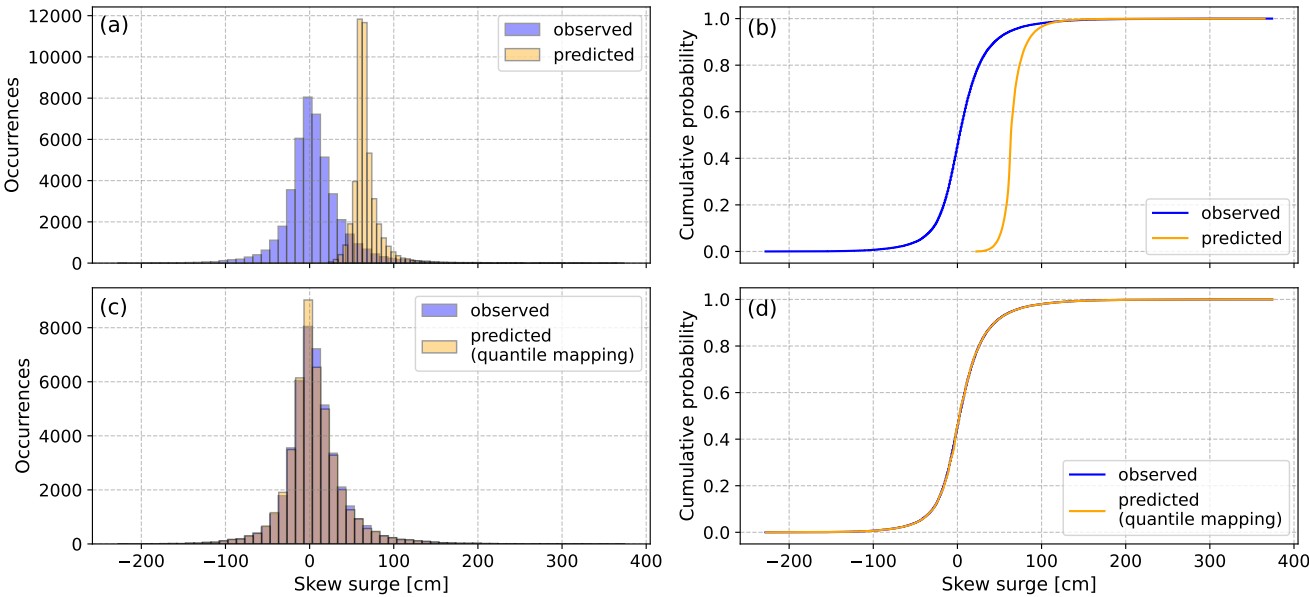

**Figure 6.** Comparison of all observed (blue) and predicted (orange) PDFs and CDFs of skew surge events from 1959 to 2022, before (a, b) and after (c, d) bias correction. The threshold for training is *hwss* $\geq$ 80 cm and the regularization method is elastic net, with the year to be predicted being excluded from the training.

Fig. 6 (a) and (b) show the probability density function (PDF) and cumulative distribution function (CDF) of the observed and predicted skew surge events from 1959 to 2022. The PDF (Fig. 6, a) reveals that the observed skew surge data has a broader distribution centered around lower skew surge heights, whereas the predicted skew surge data is more concentrated and centered around greater skew surge heights, never falling below zero. We also see the same pattern in the CDF (Fig. 6, b),

with the predicted CDF indicating higher skew surges at a given cumulative probability compared to the observed CDF.



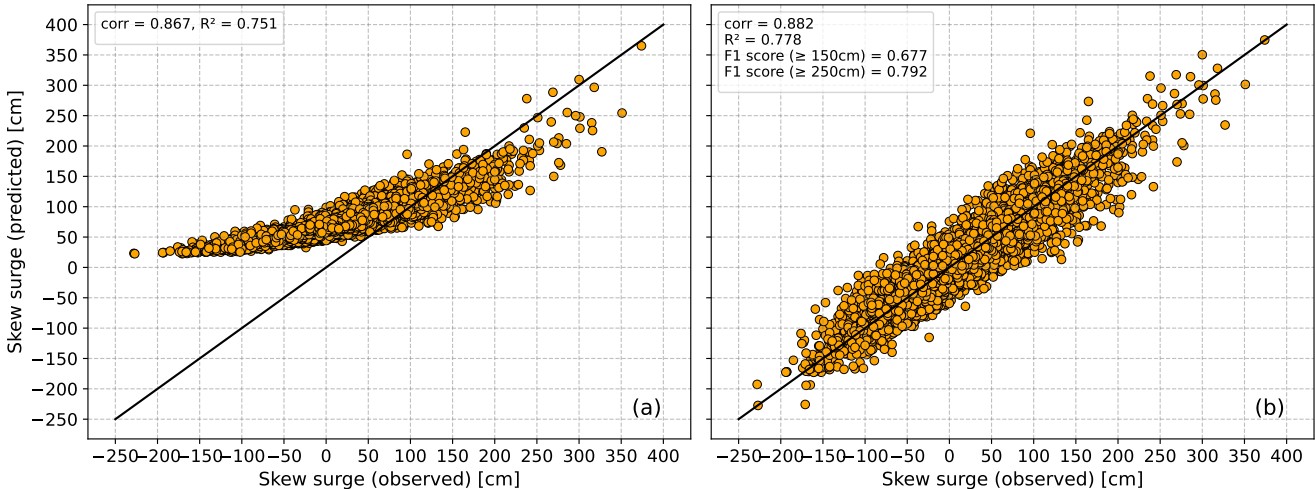

**Figure 7.** Scatter plots comparing all observed and predicted skew surge heights from 1959 to 2022, before (a) and after (b) quantile mapping bias correction. The black line represents the diagonal. The threshold for training is $hwss \geq 80$ cm and the regularization method is elastic net, with the year to be predicted being excluded from the training.

Additionally, we plot observed against predicted skew surge heights (Fig. 7, a) and find a good correlation coefficient of 0.867 and $R^2$ value of 0.751. However, in the lower range of skew surges (observed values below 80 cm), the model tends to overestimate the values, while in the upper range (observed values above 150 cm), it tends to underestimate the skew surge heights. While the underestimation of extreme skew surges may be due to a lack of data (Sect. 3.2), the overestimation of low

or negative skew surges is a consequence of training on logarithmic values. The reason for this lies in the nature of logarithmic functions which is defined only for positive real numbers.

In order to adjust the model output, we apply bias correction using the quantile mapping method of Cannon et al. (2015). Fig. 6 (c) and (d) show the PDF and CDF of the observed and bias-corrected predicted skew surge events. After applying the quantile mapping bias correction, the predicted distribution aligns much more closely with the observed distribution, both in

terms of central tendency and spread. Also, the almost overlapping CDFs (Fig. 6, d) and the tighter clustering of points around the diagonal in Fig. 7 (b) indicate that the quantile mapping method successfully corrects the bias in the model's prediction. Moreover, it effectively reduces both the overestimation of lower skew surges and the underestimation of higher skew surges, resulting in better alignment with the observed values. The corrected prediction exhibit a higher correlation with the observed values (corr = 0.882) and an increase in the $R^2$ value ($R^2 = 0.778$) (Fig. 7, b). Compared to the more complex models developed

by Jensen et al. (2013), Dangendorf et al. (2014) and Niehüser et al. (2018), our storm surge model demonstrates equally high skill measures despite its extreme simplicity.



## 3.5 Classification evaluation

To assess the ability of the storm surge model to discriminate extreme storm surge events, we conduct a classification evaluation and calculate the F1 score for two categories: 1) predicting a skew surge of greater than or equal to 150 cm, and 2) predicting 365 a skew surge of greater than or equal to 250 cm. These threshold values are officially used by the BSH to define a storm surge and a severe storm surge (Sect. 2.3.3). To perform the classification, we use values from the contingency table based on the model's prediction (Table 2).

**Table 2.** Contingency table based on the bias-corrected predictions for two categories: skew surge $\geq 150$ cm and skew surge $\geq 250$ cm.

|  | Skew surge $\geq 150$ cm | Skew surge $\geq 250$ cm |
| --- | --- | --- |
| True negatives (Correct rejection) | 44829 | 45132 |
| True positives (Hit) | 170 | 19 |
| False negatives (Miss) | 83 | 5 |
| False positives (False alarm) | 79 | 5 |

The contingency table consists of four values: true negatives, true positives, false negatives and false positives. We use the last three values to calculate precision, recall, and ultimately the F1 score, as shown in Fig. 7 (b). In our analysis, the F1 scores for 370 predicting skew surges greater than or equal to 150 cm and 250 cm are 0.677 and 0.792, respectively. This indicates good model performance at the 150 cm threshold, showing a reasonable balance between precision and recall. For the higher threshold of 250 cm, the model demonstrates even better performance. This improvement suggests that the model is more accurate and reliable in predicting larger skew surges. However, the very small sample size of just 24 events with a skew surge of more than 250 cm (19 hits and 5 misses) plus 5 false alarms is associated with much higher uncertainty for the respective F1-score.

## 4 Discussion

The presented storm surge model demonstrates a strong correlation between observed and predicted skew surges (Fig. 7, 8 a). The classification evaluation indicates that the model might be even more accurate in predicting larger skew surges ($\geq 250$ cm) compared to those above 150 cm. Other studies, such as those by Müller-Navarra and Giese (1999), Niehüser et al. (2018) and Dangendorf et al. (2014), often underestimate these higher skew surges. As previously mentioned, the sample size for 380 calculating the F1 score at the 250 cm threshold is very small, so the F1-score is associated with a much greater uncertainty. Nevertheless, the score underlines that the model is effective in predicting extreme events. It is reasonable to assume that especially severe storm surges are predominantly caused by northwesterly winds, as this pattern aligns with the predictor regions (Sect. 3.3; Fig. 3: 12 h, 6 h, 2 h and 1 h). This assumption is supported by Gerber et al. (2016), who identified, analyzed, and categorized storm surges based on atmospheric conditions. They find that severe storm surges in the German







**Figure 8.** Time series of all observed and predicted (quantile mapping) skew surges from 1959 to 2022 for the entire period (a) and a zoom into the year 1976 for better visibility of particular events in an example year, containing the highest ever recorded storm surge at Cuxhaven (b). The threshold for training is $hwss \geq 80$ cm and the regularization method is elastic net, with the year to be predicted being excluded from the training.

385    Bight occur more frequently during the North-West Type (NWT), characterized by a northwesterly flow from the northern North Sea to the German Bight. Examples of storm surges resulting from a NWT weather situation include the storm surge in February 1962 and the one at the end of January 1976 (Jochner et al., 2013; Gerber et al., 2016; Meyer and Gaslikova, 2024). In these instances, high effective winds (17 to 21 m s$^{-1}$) are recorded at all predictor locations and our storm surge model successfully predicts these storm surges (Fig. 8). Another weather situation relevant for storm surges in the German Bight is

390    the West and South-West Type (W+SWT), characterized by westerly or south-westerly winds (Gerber et al., 2016). Notable examples of storm surges resulting from W+SWT weather situations include one of the highest storm surges ever recorded at the beginning of January 1976 (Gerber et al., 2016; Meyer and Gaslikova, 2024) and the storm surge in February 2022 (Mühr





et al., 2022). Here, our model shows varying performance. On the one hand, with high effective wind speeds (18 to 23 m s$^{-1}$) at all predictor locations, the storm surge model successfully predicts the storm surge on January 3, 1976 (Fig. 8, b). On the other hand, the model underestimates the storm surge in February 2022 (Fig. 8 a). In this case, effective wind speeds ranged from 15 m s$^{-1}$ to 18 m s$^{-1}$ in the central (Fig. 3: 6 h) to southeastern North Sea (Fig. 3: 2 h and 1 h), but negative effective winds of -7 m s$^{-1}$ at the predictor location in the very northwestern North Sea (Fig. 3: 12 h). The influence of the negative effective wind in the northwestern predictor location results in a lower predicted skew surge. However, other external factors may also influence the performance of the model. The storm surge in February 2022 was not only the result of predominantly south-westerly winds but was also part of a series of storms (Mühr et al., 2022). This led to a pre-filling of the North Sea (Mühr et al., 2022), causing weaker effective winds to induce a higher skew surge. This pre-filling effect additionally explains why the model underestimates the February 2022 storm surge. Nevertheless, the resulting skew surge prediction still exceeds 150 cm, that means it is still able to identify an event exceeding this critical threshold operationally used for issuing warnings.

When considering the categories for storm surge-inducing weather situations, the tracks of the NWT and W+SWT categories overlap in certain areas (Gerber et al., 2016). Thus, we cannot conclusively state that our model is better at predicting storm surges from one category over the other. Nevertheless, as long as the triggering weather situation leads to westerly to north-westerly winds, which is mostly the case, the storm surge model is able to predict the resulting storm surge very well.

Another factor contributing to extreme water levels in the North Sea is external surges. These surges are caused by low-pressure systems over the North Atlantic and are amplified at the continental shelf (Böhme et al., 2023). External surges can cause water-level changes exceeding 1 m along the along the British, Dutch, and German coasts (Böhme et al., 2023). When external surges coincide with storm surges, they have the potential to create extreme water levels. Of the 126 external surges recorded between 1971 and 2020, 21 % occurred during or close to a storm surge event in the German Bight (Böhme et al., 2023). Notable examples of such co-occurrences include the storm surge in February 1962 and the one in December 2013 (Böhme et al., 2023). For both events, the storm surge model predicted a skew surge of more than 150 cm in December 2013 and even more than 250 cm in February 1962 (Fig. 8, a), leading to storm surge or severe storm surge warnings. However, the actual water levels were underestimated by approximately 50 cm, which roughly corresponds to the average influence of external surges on water levels in Cuxhaven during storm surges (Böhme et al., 2023).

Despite the aforementioned limitations, we find strong model performance in predicting both moderate and extreme storm surges in the German Bight.

## 5 Summary and conclusions

In this study, we developed a statistical wind-based storm surge model for the German Bight that is able to predict skew surges based solely on the effective wind. Aiming for model simplicity, we identified predictor locations across the entire North Sea, considered numerous training thresholds, and applied three different regularization methods. The resulting storm surge model comprises only five terms: the squared effective wind in certain grid cells at four lead times (12 hours, 6 hours, 2 hours, and 1 hour prior to the skew surge event) as robust predictors, along with an intercept.





We validate the model against historical data and find that it achieves high predictive accuracy, rivaling more complex models despite its simplicity. Moreover, our findings underscore the significance of wind as the primary driver of storm surges in the German Bight. The model provides reliable predictions for both moderate and extreme storm surge events, with more accurate predictions for surges preceded by westerly or northwesterly winds. In particular, the good prediction accuracy for storm surges
greater than or equal to 250 cm is a unique outcome of this study.

Furthermore, the simplicity of our storm surge model facilitates its application to climate simulations, making it a valuable tool for assessing storm surge risk in the German Bight under changing climate conditions, on top of sea level rise. Additionally, this approach is not only effective for the German Bight but is also adaptable to other coastal regions worldwide.

**Appendix A: Grid cell models**

To determine the spatial and temporal positions of the predictors ($effwind_{10\mathrm{m}}^{k,t}\,(hwss)$) within the North Sea region, we create grid cell models. We train these models and conduct a leave-one-out cross-validation separately for each threshold ($hwss \geq 50, 60...150$ cm), performed hourly up to 12 hours before the occurrence of the respective skew surge. We then compute the $R^2$ value for each grid cell model and time step. For each time step, we select the grid cell model with the highest $R^2$ value,
specifying it as the relevant grid cell for that specific time step.



**Figure A1.** $R^2$ value (based on a leave-one-out-cross-validation) of each grid cell model within the North Sea region for 12 hours before the skew surge event (top left) to 1 hour before the skew surge event (bottom right). The darker the colour, the greater the $R^2$ value. The event threshold for training is *hwss* $\geq 80$ cm.



*Data availability.* Water level data for the Cuxhaven gauge are available from the Federal Waterways and Shipping Administration (WSV). The ERA5 reanalysis products used for this study are available in the Copernicus Data Store at https://doi.org/10.24381/cds.adbb2d47 (Hersbach et al., 2023). The skew surge data used in this study is currently being prepared for publication.

*Author contributions.* LS and TK designed and developed the study. AB computed and provided the skew surge data. LS did the analysis
and created the figures. LS, JB and TK discussed the results. LS wrote the manuscript with contributions from all co-authors.

*Competing interests.* The authors declare that they have no conflict of interest.

*Acknowledgements.* We thank Daniel Krieger and Claudia Hinrichs for valuable feedback on this manuscript. This study was funded by the German Federal Ministry of Digital and Transport (BMDV) in the context of the BMDV Network of Experts. Johanna Baehr was funded by the Deutsche Forschungsgemeinschaft (DFG, German Research Foundation) under Germany's Excellence Strategy – EXC 2037
"CLICCS – Climate, Climatic Change, and Society" – project number: 390683824, contribution to the Center for Earth System Research and Sustainability (CEN) of Universität Hamburg. We thank the German Computing Center (DKRZ) for providing their computing resources.



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
