# Peer review of "Development of a wind-based storm surge model for the German Bight"

_EGUsphere, 2024_

## Author Comment (AC1)

**Response to Reviewer #2**

We sincerely thank Reviewer #2 for their thoughtful and constructive feedback on our manuscript *Development of a wind-based storm surge model for the German Bight*. The comments contributed to improve the manuscript's clarity and quality. Below, we provide a detailed point-by-point response to the concerns raised and explain how we plan to incorporate the suggestions into the revised version.

**Main comments**

**1)** The manuscript is difficult to read due to excessive over-explanation, which does not significantly enhance the understanding of the methods. To improve readability, I suggest streamlining the text by focusing on essential details in the main body and moving less critical or overly detailed sections to the appendix. For instance, Section 2.2.1 and 2.2.2 could be relocated to the appendix. In the comments below, I will highlight specific passages that can either be omitted or moved. Overall, the authors should aim to make the paper more concise and accessible.

**Response:** We thank the reviewer for the constructive feedback regarding the readability and clarity of the manuscript. We appreciate the suggestion to streamline the text and focus on essential details in the main body while relocating less critical or overly detailed sections to the appendix.
We will revise the manuscript to improve its clarity and accessibility. Specifically, we will:
1. Relocate Sections 2.2.1 and 2.2.2 to the appendix, as suggested, ensuring that the main body emphasizes the most relevant details for understanding the methods.
2. Review and revise other passages mentioned in the detailed comments, omitting redundant explanations and refining the text to avoid over-explanation.

We believe that these changes will significantly improve the readability of the manuscript and make it more concise without compromising its scientific quality.

**2)** Regarding the use of the effective wind 12 to 1 hour before the skew surge event, I understand the logic explained in the paper, but am not sure of its broader purpose. If this method is intended to detect surges in climate model simulations, wouldn't it be sufficient to use e.g., the 1-hour or 0-hour time step, unless the inclusion of 12-, 6- and 2-hour time steps (as in Equation 8) demonstrably improves model skill? Could the authors comment on the extent to which these additional time steps improve the predictive ability of the model? Furthermore, given that most model outputs are available at 3- or 6-hour intervals, wouldn't the 2- and 1-hour time steps be impractical for such applications?

**Response:** We thank the reviewer for the thoughtful feedback regarding the use of the effective wind from 12 to 1 hour before the skew surge event. We appreciate the opportunity to clarify the purpose of this approach.
1. Importance of 12 to 1-hour time steps (hourly): The inclusion of time steps from 12 to 1 hour before the skew surge event is crucial for the accurate description of the physical process underlying water elevation at the coast. Limiting the model only to

the 1-hour or 0-hour time steps would not fully capture the temporal dynamics that influence the skew surge formation.

2. Demonstrated improvement in model skill: After presenting all time steps (12 to 1 hour prior to the skew surge event) as potential predictors, our regularization approach, the elastic-net procedure, identified the 12-, 6-, 2-, and 1-hour time steps as relevant predictors in the model. The regularization procedure explicitly checks whether the inclusion of predictors significantly improves skill compared to a simpler model with less predictors. So yes, the inclusion of several time steps significantly improves the model compared to a model relying on a single time step.

3. Applicability to climate simulations with coarser resolution: It is absolutely correct that climate simulations typically provide output at 3- or 6- hour intervals, which could make the use of 2- and 1-hour time steps impractical in such applications. Still, we developed the approach based on higher temporal resolution to provide some kind of benchmark. A follow-up study will deal with coarser resolution and assess in how far results of similar quality can be achieved.

The main objective of this paper is to describe and demonstrate the developed statistical method using the best temporal resolution (hourly) available to us. This high-resolution approach allows us to present and validate the methodology in its most effective form.

**3)** The bias correction (Figures 5 and 6) does not significantly improve the model performance for values above 80 cm. If the focus is on values below 80 cm, a threshold of 50 cm or lower could be considered instead of 80 cm. Overall, this additional step might not be necessary.

**Response:** We thank the reviewer for the comment regarding the bias correction and its impact on model performance for values above 80 cm. We would like to address this point and clarify a few details:

1. Figures 5 and 7 clarification: The bias correction is not applied in Figure 5, which instead presents the results of different regularization methods. The effect of the bias correction is shown in Figure 6, Figure 7 and Figure 8.

2. Improvement of values above 80cm: While the bias correction does not change the correlation for values above 80 cm, it does improve the model's performance in terms of absolute values, as demonstrated in Figure 7 (b) and Figure 8.

3. Focus on values above 80cm: The model is not specifically focused on values below 80 cm. While the model trained with a threshold of 50 cm shows good correlation, this is primarily due to better performance at lower surge values, which are less important for the study's focus on extreme events. Choosing a threshold of 80 cm is a somewhat subjective decision. However, using this threshold we strike a balance between data availability, high threshold and model performance (lines 302-307).

We hope this clarification addresses the concerns, and we appreciate the valuable feedback on this aspect of the study.

**Other comments**

**Title:** I would suggest replace "wind-based" with statistical or something like that

**Response:** We thank the reviewer for the suggestion. While we appreciate the recommendation, we prefer to retain the term "wind-based" as it makes it clear that the model relies on a single variable – wind – as its primary input. The term "statistical" is broader and could imply a variety of approaches, including more complex or multi-variable models.

**Line 71:** replace "However, as all these" with → all mentioned

**Response:** We will apply the proposed change.

**Line 86:** remove "of relevant … positions of the"

**Response:** We will apply the proposed change.

**Line 88-89:** "(c) … training the model… (d) training and …" Are "training" in c and d different?

**Response:** We thank the reviewer for the question regarding the use of "training" in points (c) and (d). The training process itself is not different between (c) and (d). Both involve the same methodology. However, the validation approaches differ. In (c), the Leave-One-Out-Cross-Validation is applied. In (d), we use a one-year-out approach for validation (for all skew surge events). In (c) we also focus on evaluating the model for different training thresholds and select the most suitable threshold as well as regularization method.

**Line 94-95:** "...time series of high water skew surge (hwss)… 1959-2022" could you show a figure? Is it figure 8?

**Response:** Yes, we show the time series of high water skew surges (hwss) for 1959-2022 in Figure 8, represented by the blue line. We will include a reference to Figure 8 in the revised version of the manuscript.

**Figure 2:** panel (b) instead of "identifying relevant predictor position" with something like→ locating the effective wind with the highest value of ?? for every time step prior to the event

**Response:** We thank the reviewer for the suggestion to revise the title in the panel (b). While we understand the intention behind the proposed wording, we have decided to keep the title short to avoid overload and ensure readability. Although the proposed title is descriptive, it primarily explains the method, which is already described in Figure 2 and in the text (Section 2.2.1).

**Table 1:** These data points are over which period? Could you include 250 cm as well?

**Response:** The data points in Table 1 cover the entire period from 1959 to 2022. We will include this information in the revised manuscript. The listed thresholds represent those we tested and used for training the model. Selecting these thresholds involved balancing the need for sufficient number of data points and high threshold value. For a threshold of 250cm, there are only 24 datapoints, which is far too few for robust model training.

**Please improve the writing in Section 2.1.2 for clarity and readability:**

**Line 119-120:** remove "a grid cell wise". For composite analysis, normally you describe them as e.g., composite map of uas and vas based on threshold of hwss for the period of 1959-2022…

**Response:** We will apply the proposed change.

Remove details of ERA5 or mention it earlier

**Response:** We will apply the proposed change.

Remove line 127-129

**Response:** We thank the reviewer for the suggestion to remove lines 127-129. However, this sentence is necessary to describe $k$ and $t$, which are essential for understanding the equation.

Bring the equation after line 132

**Response:** We will apply the proposed change.

**Section 2.2:**

Please move Sections 2.2.1 and 2.2.2 to the appendix, retaining only the most important parts. The section could then be renamed to 'Statistical Model Development for …' or a similar title. Alternatively, if the purpose of this section is to explain the methodology, the text needs to be improved for clarity and focus.

**Response:** We thank the reviewer for the suggestion to move Sections 2.2.1 and 2.2.2 to the appendix. We agree with this recommendation and will relocate these sections to the appendix in the revised manuscript. To maintain clarity in the main text, we will keep a concise paragraph summarizing the general idea of the model development. Additionally, we will rename the section to reflect its focus on statistical model development, as suggested.

**Section 2.3:**

Most of the content can be moved to the appendix. Some information, such as the F1 score definition, is repeated later in the text.

**Response:** We thank the reviewer for the feedback on Section 2.3. While we understand the suggestion to move much of this content to the appendix, this section describes the setup of the final storm surge model, which is key element of the manuscript. Therefore, we believe it is essential to retain this section in the main text. However, we acknowledge that some details, such as the F1 score definition, RMSE, and the description of the bootstrap method, can be moved to the appendix to streamline the main text. Additionally, we will revise and summarize the subsections to improve clarity.

**Line 199-203:** repeats information already mentioned earlier in the text.

**Response:** We thank the reviewer for the observation regarding lines 199-203. We respectfully disagree that these lines repeat earlier information. These lines introduce the skew surge models, which are discussed here for the first time. We will ensure that this distinction is clear and will review the manuscript to confirm there is no redundancy.

**Line 264:** "...spatial and temporal… " please change e.g., location and time lag

**Response:** We will apply the proposed change.

**Figure 3:** would it be possible to show the same for both hwss ≥ 50 and hwss ≥ 150cm?

**Response:** We thank the reviewer for the suggestion to include additional plots for both *hwss* ≥ 50cm and *hwss* ≥ 150cm in Figure 3. While we understand the interest, we believe adding these plots would not provide significant added value to the manuscript, as for the final storm surge model we use *hwss* ≥ 80cm. For your information, we have included the requested figures below to provide additional context (Fig. R1).

[Figure]

**Figure R1:** Map of the North Sea region showing the 12 relevant grid cells highlighted with crosses. Each cross represents a specific grid cell, with the colour variations indicating the different time intervals before the skew surge event and the corresponding R2 value (based on a leave-one-out-cross-validation). The threshold for training is hwss ≥ 50 cm (upper plot) and hwss ≥ 150 cm (bottom plot).

The path of grid cells with a training threshold of *hwss* ≥ 50 cm is similar to that for *hwss* ≥ 80 cm, both following a northwest to southeast orientation (Fig. R1, upper plot). In contrast, the path of grid cells with a training threshold of *hwss* ≥ 150 cm shows no clear or consistent pattern, primarily due to the significantly lower amount of available data (Fig. R1, lower plot). Additionally, this path does not make sense in terms of the physical processes driving storm surges, further highlighting the limitations of using such a high threshold for training.

**Line 270:** remove "the grids cells in". Similarly in other part of the paper

**Response:** We will apply the proposed change.

**Line 305:** Referring to my comment 3 above, you could use 50 cm as the threshold as well, correct? From Figure 5, it's difficult to determine whether 50 cm is better than 80 cm. However, if you opt for 50 cm, would bias correction still be necessary?

**Response:** We thank the reviewer for the insightful comment. Yes, we could use 50 cm as a threshold, but we have opted for the highest possible threshold, as our aim is to develop a storm surge model. The decision of the training threshold is somewhat subjective, but it is based on the goal of optimizing its performance for extreme surge events. Moreover, Figure 5 does not show whether 50 cm is better than 80 cm, as the 50 cm threshold is not included in that figure. However, Figure 4 provides the $R^2$ and RMSE values for all thresholds, where we observe that the model trained with a 50 cm threshold performs slightly better than trained with an 80 cm threshold, though not significantly. The good correlation of the predicted skew surges (training threshold 50 cm) with the observed values is mainly due to lower values. Furthermore, bias correction would still be necessary if we were to opt for the 50 cm threshold, as extreme values would be further underestimated. Additionally, due to the logarithmic transformation, lower values would not drop below zero, making bias correction crucial for accurate predictions.

**Figure 5:** please show the chosen method (elastic net) but instead show a similar plot for both hwss ≥ 50 and hwss ≥ 150. Please also include the fitted line for the data.

**Response:** We thank the reviewer for the thoughtful suggestions regarding Figure 5. The purpose of this figure is to illustrate the differences among the three regularization methods rather than to compare thresholds. The differences between thresholds, including *hwss* ≥ 50 cm and *hwss* ≥ 150 cm, are already shown in Figure 4. While we believe adding plots similar to figure 5 but for *hwss* ≥ 50 cm and *hwss* ≥ 150 cm does not significantly enhance the manuscript, we include these figures here for your reference (Fig. R2).

[Figure]

**Figure R2**: Comparison between observed skew surge heights and predicted skew surge heights resulting from a leave-one-out-cross-validation elastic net regression. The black line represents the diagonal. The threshold for training is hwss ≥ 50 cm (a), hwss ≥ 80 cm (b) and hwss ≥ 150 cm (c).

The good correlation of the model trained with a 50 cm threshold is primarily driven by lower values and higher surges are still underestimated. When comparing the model outputs trained with thresholds of 50 cm (Fig. R2 a) and 80 cm (Fig. R2 b), the visual spread of points in the range of 200 cm to 350 cm (observed values) is smaller when using the 80 cm threshold. Again, the choice of 80 cm as the training threshold is subjective and represents a balance between data availability, high threshold and model performance (lines 302-307).

With a training threshold of *hwss* ≥ 150 cm (Fig. R2 c), the model has too few data points to train effectively. Additionally, the location and time lag of the predictors identified for this threshold do not make sense in terms of the physical processes driving storm surges. Furthermore, using this threshold does not significantly enhance the prediction of high surge values, making it less useful.

Although we understand the idea of including fitted lines in Figure 5, we have decided not to include them in the manuscript. While these lines illustrate the correlation especially for lower values, the focus of the figure is on high surge values, which are more relevant to the manuscript's objectives.

**Figure 7:** Please include F1 score for panel a plot.

**Response:** We will apply the proposed change.

**Table 2:** Please show the values without bias correction as well. Also mention the number of events from observations.

**Response:** We thank the reviewer for the comment. The number of events from observations corresponds to the sum of Hits and Misses. While we acknowledge its importance, we have decided not to include this information directly in the table to maintain its clarity and focus. Moreover, we agree with the suggestion to include the values without bias correction and will add them to the revised version of the manuscript. This will allow for a more comprehensive comparison.

**Line 376-378:** Is this statement for bias corrected case? Then you can use as an argument to my comments above.

**Response:** Yes, the statement in lines 376-378 refers to the bias-corrected case, as shown in Figures 7b and 8. This highlights the importance of applying bias correction, as it significantly improves the model's performance in terms of absolute values for higher surge values and eliminates the underestimation observed without correction. We hope this explanation clarifies the context of the statement and emphasizes the need for a bias correction in the model. Accordingly, we will include an explicit statement in this respect in the revised manuscript. We thank the reviewer for pointing this out.

---

## Author Comment (AC2)

Response to Reviewer #1

We sincerely thank Reviewer #1 for their insightful and constructive feedback on our manuscript *Development of a wind-based storm surge model for the German Bight*. The comments contributed to improving the clarity and quality of the manuscript. Below, we provide a detailed point-by-point response to the issues raised and propose how we intend to incorporate the suggestions into the revised manuscript.

**Eq.(1):** Are the mean wind speed ($U_k$) and mean zonal/meridional winds calculated from all timesteps of the reanalysis or only those related to the skew surges from the corresponding *hwss*? If all timesteps are considered, then the normalized mean wind ($uas_k/U_k$, $vas_k/U_k$) is an averaged normalized wind at this grid cell independently on whether it causes storm surge or not and thus effwind is a projection on average wind direction, not the direction favorable for the generation of surges. In this case it is unclear why it is called "effective wind", effective for what? If only winds associated with skew surge higher than a certain threshold are considered to construct the normalized mean, please specify which timesteps exactly were taken (e.g. at the moment of the skew surge or within 12 hours prior or something else)?

**Response:** We apologize for the unclear explanation regarding the calculation of the effective wind. To clarify, we perform the calculation of the mean wind speed and mean zonal/meridional winds using only the timesteps associated with skew surges that are greater or equal to the *hwss* training threshold. Moreover, we calculate the specific wind direction through a composite analysis, which is done separately for each hour (up to 12 hours) prior to every *hwss* event. We will adjust the description within the paragraph with a clearer explanation.

**L145-153:** I'm trying to understand the shape of the individual model (for each grid cell, time lag, hwss). From this passage I would assume it looks like log(skew_surge)=a*(ef_wind²) +b*(ef_wind)+c. If this is not what was used, please reconsider the description. If this is what was used, then (1) just for the sake of terminology, this is not a multiple linear regression as stated in Line145, but rather a simple quadratic regression (2) I find the sentence "In this way, we ensure the effect of the negative sign" misleading here. Firstly, for high skew surges (hwss) effective winds will be positive anyway. Yes, consideration of only positive coefficients may be helpful later when the skew surges for the whole year are reconstructed and negative effective winds are well possible, as explained in Sect. 2.3.2, but to ensure influence of negative effective winds a special treatment is necessary and positive coefficients alone do not ensure the effect of negative sign. I suggest to refer here directly to the Sect. 2.3.2 for more coherent explanation of this constrain for those interested.

**Response:** Absolutely, the individual model (for each grid cell, time lag and *hwss*) looks like log(skew_surge) = a*(eff_wind²)+b*(eff_wind)+c. We apologize for inadvertently using the wrong terminology and will change it to simple quadratic regression. We thank the reviewer for pointing this out.
Correct, to ensure the influence of negative effective winds a special treatment is necessary. We will delete the sentence "In this way, we ensure the effect of the negative sign" and refer to Sect. 2.3.2 instead.

**General comment:** I wonder how sensitive is the model to the selected training dataset? That is, how much the models (and the estimated skew surge) change when different years are excluded, as it has been done during the validation procedure? If, for example, years Y1 and Y2 are excluded to generate Model1, Y1 and Y3 are excluded to generate Model2, would the reconstructions of skew surges for the year Y1 from these two models be identical? More generally, especially if the model to be used for scenarios, does the size or selection of the training dataset matter?

**Response:** We thank the reviewer for the insightful comment regarding the sensitivity of the model to the selected training dataset. Yes, the size of the training dataset does indeed matter: If the dataset is small and includes many predictors, the model may overfit, capturing noise rather than the underlying patterns. A larger dataset provides more information, which leads to more stable and reliable coefficient estimates. In turn, this helps the model perform better on unseen data by balancing bias and variance effectively.

To answer the specific question, we conducted the proposed experiment using the year 1976. For Model 1, we excluded 1976 and 1977, while for Model 2, we excluded 1976 and 1975. The reconstructed skew surges for the year 1976 from both models were of the same order of magnitude, but not identical, indicating that the model's output can vary depending on the years excluded from the training dataset.

For scenario applications, we will train the model using all available years from 1959 to 2022, ensuring that the model is based on a comprehensive dataset. However, we would like to emphasize that we perform Leave-One-Out Cross-Validation during the validation procedure to ensure that the final model achieves maximum generalization performance and is as representative as possible.

**L20:** "Coastal protection institutions"  - Meant are those organizations who plan and construct protection structures? Is this an established term?

**Response:** Yes, we will rephrase that term to ensure clarity.

**L22:** "continuing rise of sea level" -> continuing rise of mean sea level

**Response:** We will apply the proposed change.

**L24:** "as sea level pressure" -> as atmospheric sea level pressure

**Response:** We will apply the proposed change.

**L28:** "... these events: The storm surges studied include..." -> ... these events. The storm surge studies include ...

**Response:** We will apply the proposed change.

**L36:** The most widely used and reliable method of such translation is a hydrodynamic model. It is clear, that this study is about simple fast methods beyond classical models, still I think the dynamical models should be mentioned somewhere in the text. Maybe within a short explanation why they are not always the best choice and where the alternatives are needed.

**Response:** We thank the reviewer for the suggestion. We agree that hydrodynamic models are the most widely used and reliable method for translating atmospheric conditions into storm surge estimates. To address the reviewer's comment, we will include a brief explanation in the text outlining why these models are not always the best choice and why simpler alternatives, such as the method proposed in this study, are useful for particular purposes.

**L57:** "They find that the external surge…" -> They find that thus considered external surge and…

**Response:** We will apply the proposed change.

**L64:** "excluded in the model setup" -> excluded from the model setup

**Response:** We will apply the proposed change.

**L274:** "the track consistently follows a northwest to southeast orientation" – I presume here the track refers to the path of grid points with maximum R^2 and not the storm track. Maybe choose another word because "track" has a certain connotation. The position of crosses also doesn't help to identify the storm track itself, maybe only hints to it, so the relation to the next sentence about prevailing northern storm tracks is not obvious.

**Response:** We thank the reviewer for pointing this out. Indeed, the word "track" could be misleading here. We will write "temporal evolution of grid points with maximum $R^2$" instead.

**L352-356:** On what dataset the bias correction (quantile mapping) was trained before it was applied to the omitted years? Would be helpful to specify in the text what was the training and what was the validation datasets.

**Response:** We thank the reviewer for this question. To clarify, the first step involves reconstructing all skew surges from 1959 – 2022 by training on all years except the year to be predicted. Subsequently, we apply the quantile mapping bias correction to all reconstructed skew surges from 1959 to 2022, without the use of cross-validation. We calculate the transfer function by comparing the observed skew surges with the reconstructed skew surges, and we then apply this transfer function to the reconstructed skew surges. We will ensure to include this clarification in the revised manuscript.